# Reproducibility Study of "Explaining RL Decisions with Trajectories"

**Clio Feng**
*University of Amsterdam*                                            *clio.feng@student.uva.nl*

**Colin Bot**
*University of Amsterdam*                                            *colin.bot@student.uva.nl*

**Bart Aaldering**
*University of Amsterdam*                                      *bart.aaldering@student.uva.nl*

**Bart den Boef**
*University of Amsterdam*                                      *bart.den.boef@student.uva.nl*

**Reviewed on OpenReview:** *https://openreview.net/forum?id=JQoWmeNaC2&noteId=HN2YxBATRB*

## Abstract

This paper reports on the reproducibility study on the paper *'Explaining RL Decisions with Trajectories'* by Deshmukh et al. (2023). The authors proposed a method to elucidate the decisions of an offline RL agent by attributing them to clusters of trajectories encountered during training. The original paper explored various environments and conducted a human study to gauge real-world performance. Our objective is to validate the effectiveness of their proposed approach. This paper conducted quantitative and qualitative experiments across three environments: a Grid-world, an Atari video game (Seaquest), and a continuous control task from MuJoCo (HalfCheetah). While the authors provided the code for the Grid-world environment, we re-implemented it for the Seaquest and HalfCheetah environments. This work extends the original paper by including trajectory rankings within a cluster, experimenting with alternative trajectory clustering, and expanding the human study. The results affirm the effectiveness of the method, both in its reproduction and in the additional experiments. However, the results of the human study suggest that the method's explanations are more challenging to interpret for humans in more complex environments. Our implementations can be found on GitHub.

## 1 Introduction

Reinforcement Learning (RL) demonstrates remarkable performance in dynamic settings, enabling real-time decision-making through direct engagement with the environment. However, the application of RL in practical contexts poses significant challenges. Offline RL research, which relies on pre-collected data rather than real-time interaction, attempts to solve issues of learning efficiency and environmental risks. An important issue that remains is the lack of explainability of the RL decision-making processes.

Previous studies have focused on explaining RL agents' by highlighting important aspects of observations leading to agent decisions (Gupta et al., 2019; Iyer et al., 2018; Greydanus et al., 2018). Seeking to tackle this issue from a different perspective, Deshmukh et al. (2023) introduced an approach that highlights which past experiences ('trajectories' from pre-collected data) are responsible for an RL agent's decision. This concept of 'attributing' an algorithm's decisions to training data has previously demonstrated effectiveness in supervised learning settings by Nguyen et al. (2021). In RL, trajectories are sequences of states, actions, and rewards from an environment that an agent traverses. The paper proposes a trajectory attribution solution in offline RL scenarios using sequence modelling networks.

This work aims to examine the results by reproducing and extending the experiments presented in the original paper. To assess the approach's efficacy, the original paper conducted experiments across diverse environments, including: i) Grid-world, a puzzle game where the objective is to navigate from a starting point to a winning position while avoiding obstacles, characterised by discrete state and action spaces. ii) Seaquest, an Atari game where the player controls a submarine that must shoot enemies while saving divers, without running out of oxygen. It has a continuous state space where observations are screen frames and a discrete action space (Bellemare et al., 2013). iii) HalfCheetah, a continuous control task where the goal is to make a 'cheetah' figure move forward by manipulating its joint angles, featuring continuous state and action spaces (Todorov et al., 2012). Figure 1 shows example states for all three environments.

In this paper, we successfully reproduce the explanation framework and experiments in the three environments above. To provide a comprehensive overview, Section 2 outlines its scope, and Section 3 details the proposed method and experiment implementation details. Replicated qualitative and quantitative results are discussed in Section 4.1, with additional experiments and a human study in Section 4.2. The process of reproducing the experiments is discussed in Section 5, concluding in Section 6.

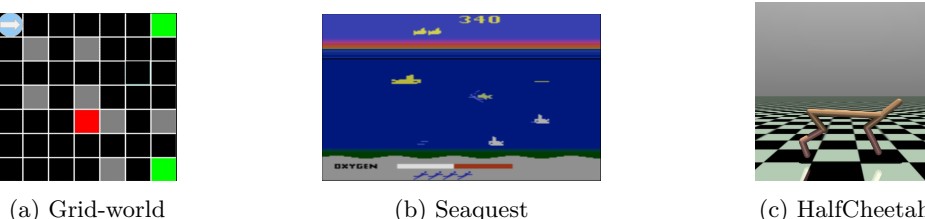

(a) Grid-world        (b) Seaquest        (c) HalfCheetah

Figure 1: **Visual representations of example states in the Grid-world, Seaquest, and HalfCheetah environments.** In the Grid-world visualization, winning states are depicted in green, while losing states are shown in red. The HalfCheetah state visualization utilized the built-in renderer.

## 2 Scope of Reproducibility

The scope of reproducibility in this paper is centered around the author's explainability framework for deep reinforcement learning. The main claim made by the author is as follows:

*'The proposed approach is efficient in terms of attribution quality and practical scalability across a spectrum of environments, including grid worlds, video games (Atari), and continuous control scenarios (MuJoCo).'*

Supporting claims and observations derived from the original paper include:

1. **Interpretable Trajectory Clustering:** Trajectory clusters generated by various suitable algorithms demonstrate consistent semantically meaningful high-level behavior.

2. **Qualitative Performance:** The method effectively explains agent actions with semantic intent by attributing relevant trajectory clusters.

3. **Quantitative Performance:** The method achieves consistent quantitative results for multiple relevant metrics outlined in Section 3.2.

4. **Human Study Insights:** Humans predominantly choose trajectories attributed by the method as the best explanation and sometimes fail to correctly identify the factors influencing an RL decision.

By reproducing the experiments and analyses conducted in the original paper for the Grid-world, Seaquest, and HalfCheetah environments, we aim to validate the main claim by affirming these Sub-claims.

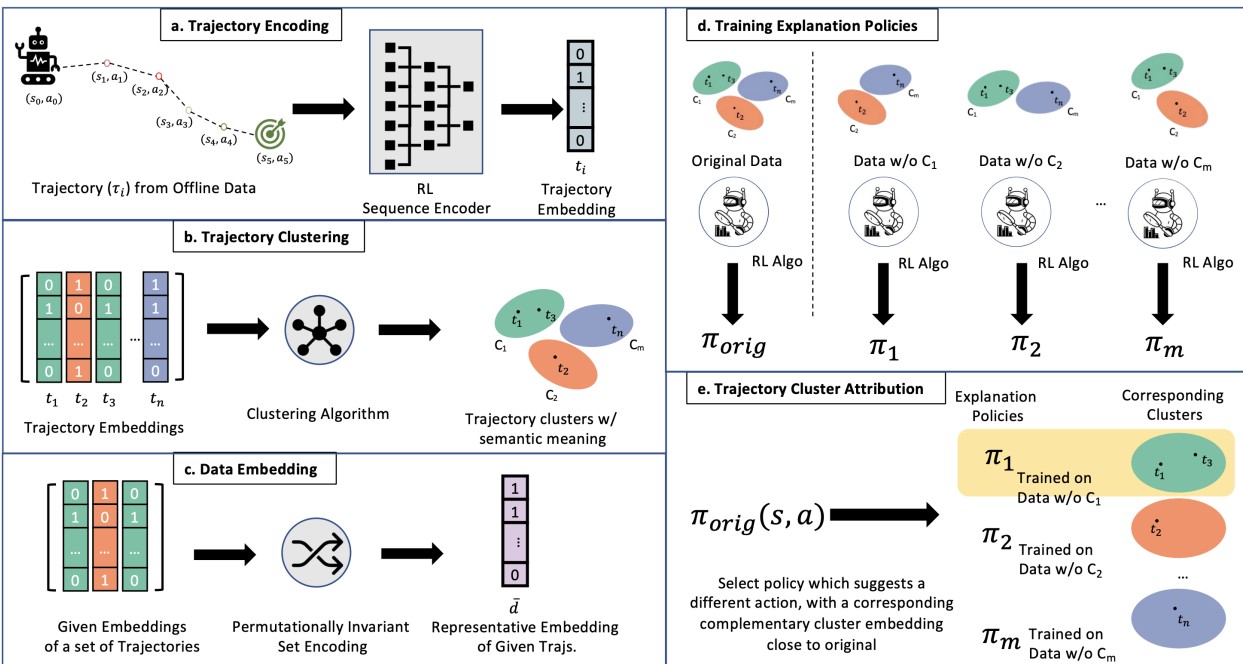

Figure 2: **Trajectory Attribution in Offline RL.** Figure from original paper Deshmukh et al. (2023)

## 3 Methodology

### 3.1 Description of methods

This section will detail the proposed trajectory attribution process illustrated in Figure 2. Initially, a set of trajectories, representing sequences of environment states, actions, and rewards, is acquired. In step $a$ each trajectory is encoded into a latent sequence using a seq2seq model, which differs for each environment. A trajectory's embedding $\tau_j$ is defined as the average of the vectors from its encoded sequence.

In step $b$, the encoded trajectories $T = \{\tau_j\}$ are grouped into clusters $C$ by a clustering algorithm, which is set to the X-means algorithm (Pelleg & Moore, 2000) by default. This algorithm automatically determines the number of clusters, identifying diverse patterns in trajectories without imposing a fixed cluster number as a hyperparameter. The original authors suggest the potential applicability of various algorithms for trajectory clustering, a claim that we examine in this paper. The explanations generated by the framework thus attribute decisions to clusters of trajectories, rather than individual trajectories. This addresses the computational constraints that come with large trajectory datasets.

In step $c$, a 'complementary dataset' is created for each cluster $c_j$ by removing it from the original dataset. For each complementary dataset as well as the entire dataset, an embedding $\bar{d}_j$ is computed. This is done by applying a softmax with temperature $T_{\text{soft}}$ to the sum of trajectories. Note that the original paper mentions dividing the sum of embeddings by a normalizing factor before applying the softmax with temperature, but that dividing by $T_{\text{soft}}$ achieves the same results.

In step $d$, an RL agent is trained on every cluster's complementary dataset, as can be seen in Figure 2d. These 'explanation policy' agents will be used for the cluster attribution. This process, along with the generation of all cluster data embeddings, is detailed in Algorithm 1.

In step $e$, actions from the original policy are attributed to a cluster by explanation policy agents. They select an action for the observed state and the algorithm chooses candidates based on the highest distances to the original agent's action. The metric for this distance is not specified in the original paper. The candidate cluster with the smallest Wasserstein distance to the original dataset's cluster embedding is selected as the responsible cluster for the state-action pair, as outlined in Algorithm 2. Essentially, this means that the

---

**Algorithm 1:** trainExpPolicies

---

**Data:** Offline Data $\{\tau_i\}$, Trajectory Embeddings $T$, Trajectory Clusters $C$, Offline RL Algorithm
    *offlineRLAlgo*

**Result:** Explanation Policies $\{\pi_j\}$, Complementary Data Embeddings $\{\bar{d}_j\}$

**for** $c_j$ *in* $C$ **do**

    $\{\tau_i\}_j \leftarrow \{\tau_i\} - c_j$ ;                 `// Compute complementary dataset corresponding to` $c_j$

    $T_j \leftarrow$ gatherTrajectoryEmbeddings$(T, \{\tau_i\}_j)$ ; `// Gather corresponding trajectory embeddings`

    Explanation policy, $\pi_j \leftarrow$ offlineRLAlgo$(\{\tau_i\}_j)$;

    Complementary data embedding, $\bar{d}_j \leftarrow$ generateDataEmbedding$(T_j, M, T_{\text{soft}})$;

**Output:** Explanation Policies $\{\pi_j\}$, Complementary Data Embeddings $\{\bar{d}_j\}$

---

algorithm identifies the cluster whose exclusion significantly alters the action while minimally affecting the data embedding.

---

**Algorithm 2:** generateClusterAttribution

---

**Data:** State $s$, Original Policy $\pi_{\text{orig}}$, Explanation Policies $\{\pi_j\}$, Original Data Embedding $\bar{d}_{\text{orig}}$,
    Complementary Data Embeddings $\{\bar{d}_j\}$

**Result:** Final Cluster Attribution $c_{\text{final}}$

Original action, $a_{\text{orig}} \leftarrow \pi_{\text{orig}}(s)$;

Actions suggested by explanation policies, $a_j \leftarrow \pi_j(s)$;

$d_{a_{\text{orig}},a_j} \leftarrow$ calcActionDistance$(a_{\text{orig}}, a_j)$ ;               `// Compute action distance`

$K \leftarrow \text{argmax}(d_{a_{\text{orig}},a_j})$ ;                `// Get candidate clusters using argmax`

$w_k \leftarrow$ Wasserstein$(\bar{d}_{\text{orig}}, \bar{d}_k)$ ;       `// Compute Wasserstein distance between` $\bar{d}_k$ `and` $\bar{d}_{\text{orig}}$

$c_{\text{final}} \leftarrow \text{argmin}(w_k)$ ;      `// Choose cluster with the minimum data embedding distance`

**Output:** $c_{\text{final}}$

---

Every environment's algorithm implementation uses a different seq2seq trajectory encoder. The Grid-world implementation uses a tokenizer to encode the states and actions as numbers, followed by a trajectory transformer as introduced by Janner et al. (2021) modified by replacing the transformer module with a Long Short-Term Memory (LSTM) cell (Hochreiter & Schmidhuber, 1997). The Seaquest environment uses a decision transformer as introduced by Chen et al. (2021). The HalfCheetah implementation uses an unmodified trajectory transformer. The deep RL agents used for the explanation policies differ per environment implementation. The Grid-world implementation uses a simple model-based algorithm using a table of transition probabilities and action values computing using the Bellman equation (Bellman, 1957). The Seaquest implementation uses Discrete Soft Actor-Critic (SAC) agents (Christodoulou, 2019). The HalfCheetah implementation uses regular SAC agents (Haarnoja et al., 2018). For Grid-world and Seaquest, a binary function determined action distance, while the actions with the top 3 Euclidean distances were chosen as candidates for HalfCheetah.

To gain an insight into the semantic meaning of a cluster, we select one trajectory that is most representative. This is done by ranking trajectories within a cluster in terms of how similar they are to the observation to be explained. By comparing all the observations in a trajectory to the observation using a similarity metric, we measure the relevance of the trajectory. This allows us to choose the most similar trajectory as the individual attribution. For Seaquest, we compare mean squared error with structural similarity, which should be more representative of similarity (Wang et al., 2004; Wang & Bovik, 2009). The HalfCheetah observations are vectors, for which the use of the Euclidean distance is satisfactory.

## 3.2 Evaluation metrics

To quantify the effects of the proposed algorithm, the authors introduced five metrics. First, shown in Equation 1, **Initial State Value Estimate** (ISVE) (Paine et al., 2020) represents the expected rewards

after completing a full trajectory, based on an initial state $s_0$ by some value function $V$. A higher ISVE indicates that a policy is trained well.

$$\mathbb{E}(V(s_0)) \tag{1}$$

The second metric is the **Local Mean Absolute Action Value Difference** (LMAAVD) in Equation 2. The action values, cumulative rewards of taking a certain action, are calculated using the action-value function $Q$. LMAAVD considers the absolute difference between actions suggested by an explanation policy $\pi_j$ for cluster $j$, as perceived by the original policy $\pi_{\mathrm{orig}}$.

$$\mathbb{E}(|\Delta Q_{\pi_{\mathrm{orig}}}|) = \mathbb{E}(|Q_{\pi_{\mathrm{orig}}}(\pi_{\mathrm{orig}}(s)) - Q_{\pi_{\mathrm{orig}}}(\pi_j(s))|) \tag{2}$$

The **Action Contrast Measure** (ACM) in Equation 3 measures the likelihood that an action suggested by $\pi_j$ is different than the action suggested by $\pi_{\mathrm{orig}}$ in state $s$. A higher ACM value for a cluster $j$ means that $\pi_j$ is likely to suggest a different action than $\pi_{\mathrm{orig}}$.

$$\mathbb{E}(\mathbb{1}(\pi_{\mathrm{orig}}(s) \neq \pi_j(s))) \tag{3}$$

The **Normalized Wasserstein Difference** (NWD), as shown in Equation 4, is a way to represent the difference between a complementary dataset embedding $\bar{d}_j$ for cluster $j$ and the original dataset embedding $\bar{d}_{\mathrm{orig}}$, normalized to the range $[0, 1]$. A lower NWD means that a complementary dataset resembles the original dataset closely, which is preferable for the candidate clusters. A lower NWD means that a complementary dataset resembles the original dataset closely. This is preferable for the candidate clusters because we want to find the cluster with the smallest change to the original dataset that leads to a different action. Intuitively, this cluster is likely to be the most responsible for the original decision.

$$W_{\mathrm{dist}}(\bar{d}_{\mathrm{orig}}, \bar{d}_j) \tag{4}$$

**Cluster Attribution Frequency** (CAF) in Equation 5 represents the probability distribution $P$ that the $j$-th cluster $c_j$ gets assigned as the responsible cluster $c_{\mathrm{final}}$ for a decision made by $\pi_{\mathrm{orig}}$.

$$P(c_{\mathrm{final}} = c_j) \tag{5}$$

### 3.3 Datasets and Models

**Grid-world.** The original paper uses 5 Dyna-Q agents placed at random start locations to obtain trajectories of lengths 1 to 15, which resulted in a dataset of 60 trajectories (Sutton, 1990). Trajectories with no final reward were ignored. 10 trajectories with a negative total reward were included as well as 50 trajectories with a positive total reward. This dataset was used to train the trajectory embedding LSTM and the agents.

**Seaquest.** We used 'seaquest-mixed-v4' from d4rl-Atari (Fu et al., 2020), as the original paper does not mention a specific dataset version. It consists of 1M pairs of observations, actions, and rewards. One observation is an 84x84 grayscale image. Corresponding to the original paper, we extracted 717 trajectories, divided into sub-trajectories of length 30. We used model weights for the decision transformer from Huggingface.

**HalfCheetah.** We used 'half-cheetah-medium-v2' from d3rlpy (Seno & Imai, 2022), as the specific version was not mentioned in the original paper. The dataset consists of 1000 trajectories of length 1000, which are divided into sub-trajectories of size 25 to make the trajectories less complex, as described in the original paper. We used model weights for the trajectory transformer from its original implementation which was trained on the same dataset.

### 3.4 Hyperparameters

**Grid-world.** The modified trajectory transformer was trained on the same dataset as the offline agents. The Dyna-Q agents were trained for 2 episodes with 5 evaluation episodes per epoch, where a training episode is one full run through the environment, with learning rates 0.1 and gamma value 0.95. The modified trajectory transformer with LSTM hidden layer size 32 was trained for 25 epochs with a learning rate 1, clipping gradients to a maximum norm of 10. The X-means algorithm was run with a cluster range between 2 and 10 clusters and $T_{\text{soft}}$ was set to 10. The offline agents had a minimum action value and transition probability of $10^{-9}$.

**Seaquest and HalfCheetah:** Seaquest's transformer ran with a vocab size of 18, a block size of 90, and 2719 timesteps, utilizing the 'reward-conditioned' model type. HalfCheetah's transformer used default parameters with a sliding window of size 10. The X-means algorithm operated with a cluster range of 2 to 8 for Seaquest and 2 to 10 for HalfCheetah. Discrete SAC and regular SAC agents from d3rlpy (Seno & Imai, 2022) were employed for Seaquest and HalfCheetah respectively, with hyperparameters consistent with the original paper: actor, critic, and temperature learning rates of $3 \times 10^{-4}$, batch size of 256 for Seaquest and 512 for HalfCheetah. $T_{\text{soft}}$ was set to $10^3$ and $10^4$ for Seaquest and HalfCheetah respectively.

### 3.5 Experimental setup

To gain insight into the amount of variability in the results, we conducted five runs with different random seeds for each of the environments and reported the means and standard deviations. The clustering was kept the same for each run because otherwise cluster labels would be shuffled.

**Grid-world.** We trained the RL policies until convergence, defined by maximum changes between iterations with a threshold of $10^{-4}$. We calculate the metrics for each agent and attribute individual trajectories from the responsible clusters. Additionally, we experimented with different clustering algorithms instead of X-means to verify their impact on the results.

**Seaquest and HalfCheetah.** We trained the RL policies for 10 epochs of $10^4$ steps. We compute the expected action values for each action for 1000 random observations with the original policy and all of the explanation policies. Using this information we attribute responsible clusters and calculate the metrics. After that, we attribute individual trajectories from the responsible clusters. To evaluate the clustering effectiveness, we compare the Principal Component Analysis (PCA) of the clustered embeddings to the original paper. We also do qualitative and quantitative analyses of the trajectory attributions.

**Human study**. Similarly to the original paper, we included two types of questions where the participant needs to select trajectories they think best explain an RL agent's action given an observation. Question Type 1 has two options with one being correct. Question Type 2 has four options with two being correct. To extend the original study, we broadened the scope to include Seaquest and HalfCheetah, increased the number of participants from 10 to 18 who had a good understanding of the RL field, and added two new question types. Question Type 3 asks the participants to rank trajectories belonging to the same cluster by relevance. Question Type 4 is an open-ended question, allowing the participant to explain previous answers or clarify what is unclear for each question. Lastly, we increased the number of questions for each type of questions per environment from 3 to 6 (for Type 1 and 2). Example questions can be found in Figure 3.

### 3.6 Computational requirements

The computational requirements are listed in Table 1. Carbon emissions are included and calculated using Machine Learning Impact calculator (Lacoste et al., 2019). Experiments were conducted using a private infrastructure, which has a carbon efficiency of 0.432 kgCO$_2$eq/kWh. The scale of our experiments was relatively small so the environmental impact was minimal.

This is an instance of the task where the agent is located at (1, 1) and is taking the 'right' action.

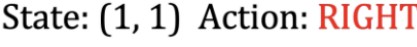

(a) Responsible Trajectory

Please explain what makes you consider your choice best explains the action suggested in a grid cell. if you do not know, why is it unclear?

(b) Type 1 & 4 question

(c) Type 2 & 4 question

Please explain what makes you consider your choices are relevant in explain the action. If you do not know, why is it unclear?

(d) Type 3 & 4 question

Mark only one oval per row.

Please explain your reasoning. If you do not know, why is it unclear?

Figure 3: **Human Survey Question Examples** for the Grid-world part of our extended human study.

Table 1: **Computational requirements**. Environment-specific requirements are listed, as well as the estimated $kgCO_2eq$ Emissions. Estimations were calculated using MachineLearning Impact calculator (Lacoste et al., 2019).

| Spec | Gridworld | Seaquest | HalfCheetah |
|---|---|---|---|
| Ran on | Jupyter Notebook | Python script | Python script |
| OS | 64-bit Ubuntu 22.04 | Windows 11 Pro | 64-bit Ubuntu 22.04 |
| CPU | 6-core Ryzen 4500u at 2.3 GHz | Intel Core i5-12400F at 4.4 GHz | 6-core Ryzen 4500u at 2.3 GHz |
| GPU | Radeon Graphics | NVIDIA GeForce GTX 960 | Radeon Graphics |
| RAM | 16GB | 16GB | 16GB |
| Total compute time (h) | 1 | 20 | 40 |
| $kgCO_2eq$ Emissions | 0.13 | 1.56 | 2.59 |

## 4 Results

### 4.1 Results reproducing original paper

In this section, we reproduce the quantitative and qualitative results from the original paper along with an extension of the original human study detailed in Section 4.2.2.

#### 4.1.1 Effectiveness of trajectory embedding & cluster generation

We attempted to prove that our reproduction of the trajectory embedding and cluster generation works similarly to the original paper's implementation. To achieve this, we generated the same figures as the original paper displaying a 2-D PCA of the trajectory embeddings per cluster. The plots for Seaquest and HalfCheetah are shown in Figure 4. Both environments show similar separation of clusters and PCA value ranges to the plots in the original paper. When inspecting the clusters by hand, they appear to encompass similar semantic behaviour (e.g. 'lining up the player with an enemy' for Seaquest and 'landing from a high jump' for HalfCheetah). The Grid-world PCA plot generated by the provided code is identical to the original paper, which is shown in Appendix 5b. These observations support Sub-claim 1 made in Section 2.

#### 4.1.2 Qualitative analysis of trajectory attributions

All three environment implementations behave similarly to the observations made in the original paper. The attributed trajectory clusters include trajectories that are both semantically related and distant to the observation that is to be explained. As mentioned in Section 4.1.1, clusters tend to contain trajectories with a similar semantic meaning. These observations support Sub-claim 2 made in Section 2. For example, an attributed cluster from the Grid-world implementation may include trajectories that pass through the same state that is to be explained, as well as trajectories that don't. Similar observations were made for the Seaquest and HalfCheetah implementations.

#### 4.1.3 Quantitative analysis of trajectory attributions

The quantitative experiments measured the metrics described in Section 3.2 for the three different environments' implementations. The code for the Grid-world experiments provided by the authors successfully reproduced quantitative results, see Appendix A.2. The quantitative results of the Seaquest experiments are shown in Table 2a. These results are different from those in the original paper. The ISVE and LMAAVD metrics mentioned in Section 3.2 are lower in all policies suggesting that they are poorly trained. The use of a reward scaler in the agents of the original paper could explain the large differences in ISVE values. However, there is no way to verify this without accessing the code of the original authors. In addition, one of the clusters' complementary datasets has a NWD of 0, which is caused by the normalization process from

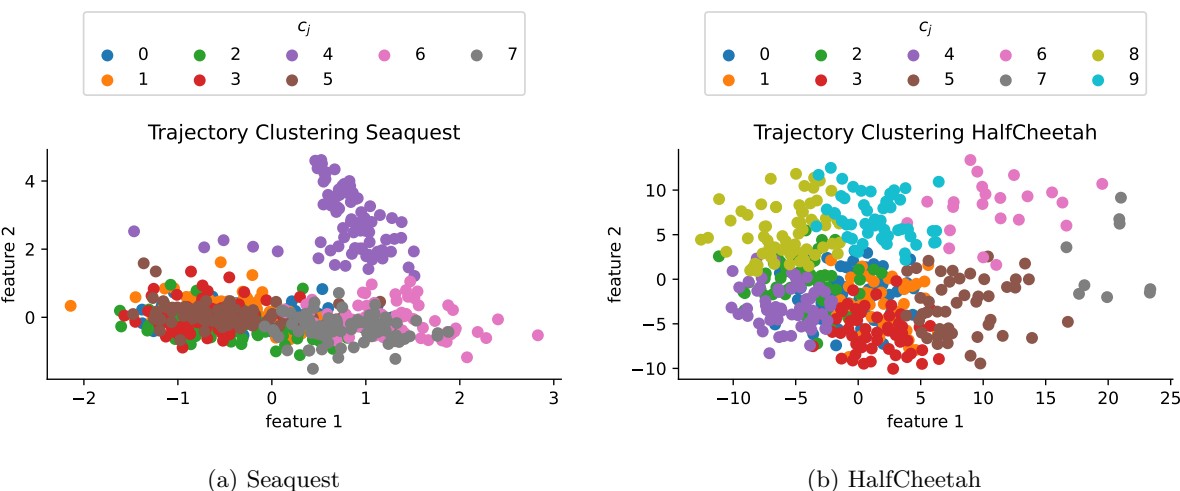

(a) Seaquest

(b) HalfCheetah

Figure 4: **Principal Component Analysis (PCA) plot depicting Clusters of Trajectory Embeddings for Seaquest and HalfCheetah.** These clusters represent semantically meaningful high-level behaviors and the trajectory embeddings are similar to those in the original paper.

the original paper. The non-normalized Wasserstein distances are not zero. As a result of this, the policy trained on this complementary dataset always gets picked whenever it has a different action, in the case of discrete action spaces. This undermines Sub-claim 1 discussed in Section 2 since one cluster can not always be the relevant one. Sub-claim 3 does hold since we were able to achieve quantitative results for the relevant metrics. The quantitative results of the HalfCheetah experiments are shown in Table 2b. The results are comparable to the ones presented in the original paper for all metrics. This also supports Sub-claim 3 in Section 2.

## 4.2 Results beyond original paper

To further test the claim of the original paper, we performed additional experiments. The results will be discussed in the section below.

### 4.2.1 Ranking Trajectories Within a Cluster

The use of mean squared error for Seaquest resulted in trajectories that were not consistently similar to the given observation, which is in line with the findings of Wang & Bovik (2009). However, we found that Structural Similarity provided an accurate ranking of the similarity in the trajectories, agreeing with Wang et al. (2004). We find that trajectories suggested through the Euclidean distance metric in the HalfCheetah implementation returned trajectories that look similar to the trajectory leading up to the observation that is to be explained, more so than returning a random trajectory from the attributed cluster.

### 4.2.2 Human study results

Analysis of the Type 1 questions reveals that in simpler environments, participants predominantly favour trajectories identified by our algorithm as the optimal explanation for the agent's actions, supporting Sub-claim 4 in Section 2. However, as environmental complexity increases, participants struggle to discern the factors influencing the agent's decision, unlike our algorithm. The accuracies for question type 1 are 77.92%, 71.48%, and 29.55% for Grid-world, Seaquest, and HalfCheetah respectively.

---

[1]Shuffling the clusters is not feasible as this would also shuffle the policies thus making the averages per index include different clusters.

Table 2: **Quantative analysis of Seaquest & HalfCheetah Trajectory Attribution.** A higher Initial State Value Estimate (ISVE) means a better-trained policy. Higher Local Mean Absolute Action Value Difference (LMAAVD) and Action Contrast Measure (ACM) mean that explanation policies suggest more contrasting actions. Normalized Wasserstein Distance (NWD) represents the difference between the complementary and original dataset of the given cluster. The Cluster Attribution Frequency (CAF) measures how often each cluster gets recognized as the responsible cluster. Clusters with low NWD and high ACM and LMAAVD are desirable. For each metric, the highest scoring cluster is denoted in **bold**. The means and standard deviations are calculated from five runs. Standard deviations of 0.000 have been denoted as '—'. Note the lack of standard deviations for the NWD metric, there was no difference between experiment runs because clusters were kept constant [1].

Table 2a: Quantitative analysis results for the Seaquest environment implementation.

| $\pi$ | ISVE $\mathbb{E}(V(s_0))$ | LMAAVD $\mathbb{E}(\|\Delta Q_{\pi_{\mathrm{orig}}}\|)$ | ACM $\mathbb{E}(\mathbb{1}(\pi_{\mathrm{orig}}(s) \neq \pi_j(s))$ | NWD $W_{\mathrm{dist}}(\bar{d}, \bar{d}_j)$ | CAF $\mathbb{P}(c_{\mathrm{final}} = c_j)$ |
|---|---|---|---|---|---|
| orig | $1.8286 \pm$ — | - | - | - | - |
| 0 | $1.9277 \pm$ — | $0.3790 \pm$ — | $0.9240 \pm$ — | $0.7831$ | $0.0030 \pm$ — |
| 1 | $1.9551 \pm$ — | $0.3818 \pm$ — | $0.9330 \pm$ — | $0.2261$ | $0.0010 \pm$ — |
| 2 | $1.8403 \pm$ — | $0.3897 \pm$ — | $0.9330 \pm$ — | $0.0000$ | $\mathbf{0.9329} \pm$ — |
| 3 | $1.9595 \pm$ — | $0.3822 \pm$ — | $0.9320 \pm$ — | $0.0406$ | $0.0090 \pm$ — |
| 4 | $1.9703 \pm$ — | $\mathbf{0.4047} \pm$ — | $\mathbf{0.9440} \pm$ — | $\mathbf{1.0000}$ | $0.0000 \pm$ — |
| 5 | $\mathbf{1.9964} \pm$ — | $0.3721 \pm$ — | $0.9160 \pm$ — | $0.2368$ | $0.0000 \pm$ — |
| 6 | $1.9421 \pm$ — | $0.3835 \pm$ — | $0.9270 \pm$ — | $0.8785$ | $0.0000 \pm$ — |
| 7 | $1.7458 \pm$ — | $0.4029 \pm$ — | $0.9320 \pm$ — | $0.0358$ | $0.0541 \pm$ — |

Table 2b: Quantitative analysis results for the HalfCheetah environment implementation.

| $\pi$ | ISVE $\mathbb{E}(V(s_0))$ | LMAAVD $\mathbb{E}(\|\Delta Q_{\pi_{\mathrm{orig}}}\|)$ | ACM $\mathbb{E}(\mathbb{1}(\pi_{\mathrm{orig}}(s) \neq \pi_j(s))$ | NWD $W_{\mathrm{dist}}(\bar{d}, \bar{d}_j)$ | CAF $\mathbb{P}(c_{\mathrm{final}} = c_j)$ |
|---|---|---|---|---|---|
| orig | $151.222 \pm 6.2520$ | - | - | - | - |
| 0 | $158.8762 \pm 7.3547$ | $0.7462 \pm 0.0892$ | $0.8527 \pm 0.0751$ | $0.5290$ | $0.0048 \pm 0.0057$ |
| 1 | $158.3154 \pm 5.4562$ | $0.6234 \pm 0.1238$ | $0.7296 \pm 0.1428$ | $\mathbf{1.0000}$ | $0.0000\pm$ — |
| 2 | $189.3018 \pm 8.3362$ | $0.7567 \pm 0.1399$ | $0.7888 \pm 0.0907$ | $0.3199$ | $0.1434 \pm 0.0430$ |
| 3 | $159.4749 \pm 9.4689$ | $0.6945 \pm 0.0925$ | $0.7537 \pm 0.1102$ | $0.4184$ | $0.0146 \pm 0.0140$ |
| 4 | $\mathbf{203.7592} \pm 21.2017$ | $0.7509 \pm 0.1129$ | $0.8067 \pm 0.0556$ | $0.9267$ | $0.0000\pm$ — |
| 5 | $122.0764 \pm 3.2605$ | $1.4960 \pm 0.3245$ | $\mathbf{1.2818} \pm 0.2074$ | $0.3910$ | $0.1194 \pm 0.0369$ |
| 6 | $162.9265 \pm 21.6331$ | $\mathbf{1.5264} \pm 0.6830$ | $1.1268 \pm 0.2711$ | $0.1901$ | $\mathbf{0.3502} \pm 0.0922$ |
| 7 | $147.2848 \pm 4.0694$ | $0.6449 \pm 0.0636$ | $0.6890 \pm 0.0789$ | $0.0000$ | $0.1678 \pm 0.0557$ |
| 8 | $178.0963 \pm 4.0394$ | $0.6301 \pm 0.0249$ | $0.7031 \pm 0.0308$ | $0.3480$ | $0.0896 \pm 0.0204$ |
| 9 | $158.9716 \pm 10.1533$ | $0.6606 \pm 0.2046$ | $0.7106 \pm 0.1449$ | $0.2130$ | $0.1102 \pm 0.0421$ |

In response to Type 2 questions, it becomes evident that not all trajectories generated by our algorithm are deemed relevant by humans; some are rated as no better than random, further confirming Sub-claim 4. Specifically, 76.56%, 66.95%, and 36.1% of participants selected trajectories generated by our algorithm for Grid-world, Seaquest, and HalfCheetah, respectively. Moreover, the accuracy decay follows a similar trend to the one observed in Type 1 questions.

To further investigate the factors that humans consider relevant, we look at the results of Type 3 questions. People tend to prioritize trajectories with the shortest path to the goal in Grid-world and visually similar trajectories for Seaquest and HalfCheetah, ranking them highest. Type 4 question responses support these observations. In Grid-world, 90% of participants cited 'reaching the goal state', 80% mentioned 'shorter trajectories', and 20% referenced 'exhibiting similar actions' as influential factors in their decisions.

### 4.2.3 Alternative Trajectory Clustering

To validate Sub-claim 1 in Section 2 beyond the X-means algorithm, alternative clustering methods were investigated. The goal is to assess if these alternatives, when applied to Grid-world trajectory data, generate similar clusters and high-level semantic behaviour.

If the clustering made by a potential algorithm is topologically and semantically similar to X-means, the algorithm is suitable for use in the framework. To validate the similarity, we used two metrics: Self-Organizing Map (SOM) (Kohonen, 1990) and Normalized Mutual Information (NMI). SOM is valuable for preserving the topology of high-dimensional data and is particularly effective in capturing spatial relationships. NMI quantifies the shared information between true and predicted clustering assignments. Both metrics yield scores between 0 and 1. A score of 1 indicates perfect agreement and a score of 0 indicates no mutual information, comparable to random chance. If a clustering scores high on both metrics, it is suitable for use in the framework.

Table 3: **Clustering Evaluation Results.** Self-Organizing Map (SOM), Normalized Mutual Information (NMI), and number of clusters compared to X-means for various clustering algorithms applied to the Grid-world trajectory dataset. More information on the tested algorithms is listed in Appendix A.3

| Method | X-means | K-means | K-medians | Dbscan | Agglo | Optic | Claran | Cure | Rock |
|---|---|---|---|---|---|---|---|---|---|
| SOM | - | 0.819 | 0.819 | 0.561 | 0.851 | 0.561 | 0.879 | 0.851 | 0.842 |
| NMI | - | 0.924 | 0.924 | 0.761 | 0.942 | 0.761 | 0.891 | 0.942 | 0.933 |
| # Clusters | 10 | 10 | 10 | 4 | 10 | 4 | 6 | 10 | 8 |

Looking at the two metrics in Table 3, it is clear that the clusters generated by most algorithms are similar to X-means. This supports the claim made by the author that there are alternative clustering methods that could effectively capture similar high-level semantic behaviour in trajectories.

## 5 Discussion

### 5.1 Reproducibility experiment experience

**What was easy**. The code was not publicly available when the reproducibility study was conducted, so contact was made with the authors to request the implementation code. They kindly provided the Grid-world experiment code on short notice. We thank the authors for their fast response and clear communication. The provided codebase produced the same results as presented in the paper. The authors provided references to pretrained decision and trajectory transformers for the Seaquest and HalfCheetah environments respectively. The authors mentioned libraries used in their implementations. The proposed algorithm is intuitive, which helps with reproducibility.

**What was difficult**. Version numbers for the libraries utilized in the experiments are missing from the original paper. This resulted in dependency issues which took time to solve. It is not entirely clear from the paper how metrics such as the ISVE and action distance were calculated for the Seaquest and HalfCheetah environments. The provided Grid-world code didn't help with clarity as the state and action spaces are different to the other two environments. This made comparing quantitative results difficult. To explore differences between ISVE values in the Seaquest environment with the original paper, we trained Discrete SAC agents with a multiplication reward scaler set to 40. This resulted in action values in the range of $[0.1691, 70.4030]$, which is much closer to the values from the original paper. Therefore, we suspect that the original paper used an unmentioned reward scaler in the Seaquest environment. No softmax temperature values were disclosed in the original paper. For Seaquest and HalfCheetah, the softmax inputs were very large, so a high temperature was needed to return softmax outputs other than 0 and 1. In the original paper no experimental reruns were reported. Thus, the robustness of the quantitative results presented in the original paper is uncertain.

## 5.2 Utility of explanations

The qualitative analysis and human study showed that the explanations generated by the proposed method are only interpretable to humans if the trajectories themselves are interpretable. As environment complexity increases, it becomes difficult to distinguish trajectories from one another. For example, the HalfCheetah environment trajectories tend to look similar. This means even though the method might attribute the perfect trajectory cluster to an action, the explanation can still be unclear. This is due to the inherent difficulty of understanding RL trajectories for humans in complex environments like HalfCheetah.

## 6 Conclusion

This paper has demonstrated our successful reproduction of the study on '*Explaining RL decisions with trajectories*' (Deshmukh et al., 2023). Despite implementation difficulties, in Section 4 we achieved results that generally align with the original work qualitatively and quantitatively, achieving interpretable trajectory clustering. In addition to this, we build upon the work in the following ways. In Section 4.2.1 we proposed methods to rank individual trajectories within a cluster to get a clearer representation of the framework's explanations. In Section 4.2.2 the human study was extended. The results show that more complex environments lead to significantly worse interpretability for humans. In Section 4.2.3 alternative clustering methods were tested, which showed that multiple clustering algorithms are suitable for the explanation framework. The results were made more robust by running the experiment multiple times.

In general, our results reaffirm the authors' claim that the approach is efficient regarding attribution quality and practical scalability. However, trajectory interpretability is still an important factor in the usefulness of the explanations. For future work, the human study results from Section 4.2.2 suggest that improvements in terms of the explanations' interpretability could be made to increase the usefulness of the explanation framework. For more complex environments, the trajectory embeddings and corresponding behaviour patterns could be analysed further, leading to more semantically meaningful clusters. This would improve the interpretability of the explanations because the clusters would be more distinct.

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

# A Appendix

## A.1 Evaluation Metrics

A clear overview of the evaluation metrics used for quantitative analysis can be found in Table 4

Table 4: **Metrics used for quantative analysis**. In these formulas, $V$ represents a value function, that returns a reward based on an input state $s$. $s_0$ represents state 0 of a given trajectory; the initial state. $Q$ is an action-value function that returns the estimated cumulative reward of taking an action decided by a policy, either the original policy $\pi_{\mathrm{orig}}$ or an explanation policy $\pi_j$ of cluster $j$, in state $s$. $W_{\mathrm{dist}}$ is a function that returns the Wasserstein distance between its inputs, which are the original dataset embedding $\bar{d}$ and the complementary dataset embedding $\bar{d}_j$ of cluster $j$. Finally, $P(c_{\mathrm{final}} = c_j)$ denotes the probability distribution that the $j$-th cluster $c_j$ is the attributed cluster $c_{\mathrm{final}}$ for a given decision.

| Abbreviation | Name | Formula |
|---|---|---|
| ISVE | Initial State Value Estimate | $\mathbb{E}(V(s_0))$ |
| LMAAVD | Local Mean Absolute Action-Value Difference | $\mathbb{E}(\|\Delta Q_{\pi_{\mathrm{orig}}}\|) = \mathbb{E}(\|Q_{\pi_{\mathrm{orig}}}(\pi_{\mathrm{orig}}(s)) - Q_{\pi_{\mathrm{orig}}}(\pi_j(s))\|)$ |
| ACM | Action Contrast Measure | $\mathbb{E}(\mathbb{1}(\pi_{\mathrm{orig}}(s) \neq \pi_j(s)))$ |
| NWD | Normalized Wasserstein Distance | $W_{\mathrm{dist}}(\bar{d}, \bar{d}_j)$ |
| CAF | Cluster Attribution Frequency | $P(c_{\mathrm{final}} = c_j)$ |

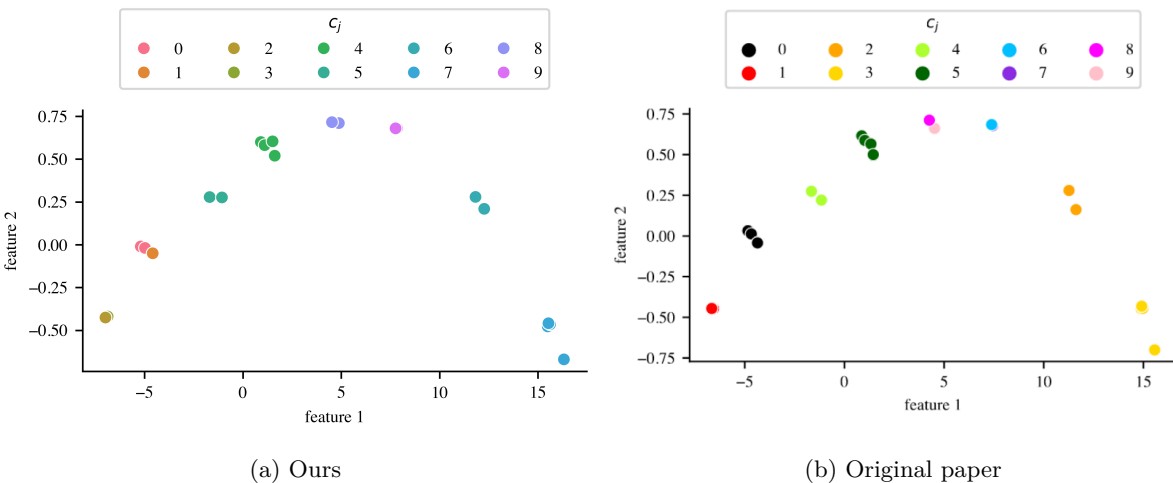

(a) Ours          (b) Original paper

Figure 5: **PCA Plot depicting Clusters of Trajectory Embeddings** of the Gridworld environment for a) our results and b) the original paper. The clusterings are almost identical to the ones presented in the original paper.

## A.2   Gridworld Trajectory Attribution Results

The results of the Gridworld trajectory attribution were very similar to the original since we were provided with the original code by the authors. Therefore, we chose to present the results of Table 5 here.

Table 5: **Quantative analysis of Gridworld Trajectory Attribution.** A higher Initial State Value Estimate (ISVE) means a better-trained policy. Higher Local Mean Absolute Action Value Difference (LMAAVD) and Action Contrast Measure (ACM) mean that explanation policies suggest more contrasting actions. Normalized Wasserstein Distance (NWD) represents the difference between the complementary and original dataset of the given cluster. The Cluster Attribution Frequency (CAF) is a measure of how often each cluster gets recognized as the responsible cluster. Clusters with low NDD and high ACM and LMAAVD are desirable. For each metric, the highest scoring cluster is denoted in **bold**. The means and standard deviations are calculated from five runs. Standard deviations of 0.000 have been denoted as '—'.

| $\pi$ | ISVE $\mathbb{E}(V(s_0))$ | LMAAVD $\mathbb{E}(\lvert\Delta Q_{\pi_{\mathrm{orig}}}\rvert)$ | ACM $\mathbb{E}(\mathbb{1}(\pi_{\mathrm{orig}}(s) \neq \pi_j(s))$ | NWD $W_{\mathrm{dist}}(\bar{d}, \bar{d}_j)$ | CAF $\mathbb{P}(c_{\mathrm{final}} = c_j)$ |
|---|---|---|---|---|---|
| orig | **0.3061** $\pm$ — | - | - | - | - |
| 0 | 0.2990 $\pm$ — | 0.0313 $\pm$ — | 0.0408 $\pm$ — | 0.0009 $\pm$ — | 0.2000 $\pm$ — |
| 1 | 0.3053 $\pm$ — | **0.0395** $\pm$ — | 0.0408 $\pm$ — | 0.0020 $\pm$ — | 0.0000 $\pm$ — |
| 2 | 0.3049 $\pm$ — | 0.0309 $\pm$ — | 0.1224 $\pm$ — | 0.0001 $\pm$ — | **0.8000** $\pm$ — |
| 3 | 0.3055 $\pm$ — | 0.0015 $\pm$ — | 0.0204 $\pm$ — | **1.0000** $\pm$ — | 0.0000 $\pm$ — |
| 4 | 0.3054 $\pm$ — | 0.0224 $\pm$ — | 0.1224 $\pm$ — | 0.0428 $\pm$ — | 0.0000 $\pm$ — |
| 5 | 0.3057 $\pm$ — | 0.0275 $\pm$ — | 0.0204 $\pm$ — | 0.0011 $\pm$ — | 0.0000 $\pm$ — |
| 6 | 0.3046 $\pm$ — | 0.0137 $\pm$ — | 0.1224 $\pm$ — | 0.0008 $\pm$ — | 0.0000 $\pm$ — |
| 7 | 0.3055 $\pm$ — | 0.0119 $\pm$ — | 0.0204 $\pm$ — | 0.0003 $\pm$ — | 0.0000 $\pm$ — |
| 8 | 0.3057 $\pm$ — | 0.0008 $\pm$ — | 0.0204 $\pm$ — | 0.0003 $\pm$ — | 0.0000 $\pm$ — |
| 9 | 0.3046 $\pm$ — | 0.0291 $\pm$ — | **0.1428** $\pm$ — | 0.0005 $\pm$ — | 0.0000 $\pm$ — |

In addition to the similar results of Table 5, the clusters found are also identical to those of the original paper. For that reason, we present Figure 5 here.

### A.3   Clustering Algorithm Information

Table 6 lists the sources for the different clustering algorithms.

Table 6: **Clustering algorithm sources**.

| Algorithm | Source |
|---|---|
| K-Means | (MacQueen et al., 1967) |
| K-Medians | (Jain & Dubes, 1988) |
| Dbscan | (Ester et al., 1996) |
| Agglo | (Jain & Dubes, 1988) |
| Optic | (Ankerst et al., 1999) |
| Claran | (Ng & Han, 2002) |
| Cure | (Guha et al., 1998) |
| Rock | (Guha et al., 2000) |

