# OpenReview forum: "Reproducibility Study of "Explaining RL Decisions with Trajectories""
_TMLR — Accepted by TMLR_

### Review · Reviewer_4piC · 2024-02-22

**Summary Of Contributions:**

The paper provides a comprehensive examination of the original work by Deshmukh et al. (2023), which introduced a method for explaining reinforcement learning (RL) decisions through trajectory clustering. This reproducibility study extends the original research by including additional experiments, extending the human study, and experimenting with alternative clustering methods.

This reproducibility study significantly contributes to the field by not only validating the original findings but also extending them in meaningful ways. It underscores the potential of trajectory-based explanations in RL while highlighting the challenges in making these explanations accessible and interpretable to a broader audience. The careful and detailed methodology, along with thoughtful analysis and suggestions for future research, make this paper a valuable resource for researchers interested in explainable AI.

**Audience:**

Yes

**Broader Impact Concerns:**

The paper poses no ethical concerns

**Claims And Evidence:**

Yes

**Requested Changes:**

## Required Changes

The paper does not require any further changes for acceptance.

***

## Recommendations to Strengthen the Paper

If desired, the following points can further strengthen the paper.

### 1. Clarification and Expansion of Methodological Details

- **Metric Calculation Clarification:** The methodology section should include a clearer explanation of how the metrics such as ISVE and action distance were calculated, especially for the environments like Seaquest and HalfCheetah, where the state and action spaces differ significantly from Grid-world.

### 2. Enhanced Discussion on Limitations and Challenges

- **Interpretability in Complex Environments:** The paper could delve deeper into the challenges of interpretability in complex environments. Suggestions for making the trajectory explanations more accessible to users without specific domain expertise would be valuable.

- **Dependence on Clustering Quality:** Discuss the potential limitations and implications of the dependence on trajectory clustering quality. This should include a discussion on how different clustering methods might impact the utility and scalability of the proposed explanation framework across various RL tasks.

### 3. Additional Experiments and Analysis

- **Robustness to Clustering Methods:** Conduct further experiments with a broader range of clustering algorithms to assess their impact on the quality and interpretability of the explanations. This could help in identifying more robust and generalizable approaches to trajectory clustering within the context of RL decision explanation.

- **Automated Evaluation of Explanations:** Explore and possibly develop automated metrics for evaluating the quality and utility of the generated explanations. This could provide a more objective basis for comparing different explanation methods in RL and facilitate a deeper understanding of what constitutes a "good" explanation.

### 4. Revision of Human Study

- **Broaden Participant Diversity:** To ensure the human study results are generalizable, consider including participants with varying levels of expertise in RL, not just those who have a good understanding of the field. This could provide insights into how intuitive the explanations are to a broader audience.

- **Detailed Analysis of Human Study Results:** Expand the analysis of the human study results to explore why participants struggled with interpreting the explanations in more complex environments. This could involve a qualitative analysis of the open-ended responses to understand the specific aspects of the explanations that were difficult to interpret.

### 5. Environmental Impact

- **Detailed Discussion on Computational Efficiency:** While the paper mentions the computational requirements and carbon emissions, it would be beneficial to discuss strategies for minimizing the environmental impact of such studies. This could include recommendations for more efficient computational practices in ML research.

### 6. Conclusion and Future Work

- **Explicit Recommendations for Future Research:** The conclusion should include explicit recommendations for future research directions, based on the findings and limitations of the current study. This could involve suggestions for alternative models or methods for explaining RL decisions, as well as potential improvements to the existing framework to enhance its interpretability and applicability.

**Strengths And Weaknesses:**

## Strengths

- **Comprehensive Methodology:** The paper carefully details the methods used to reproduce and extend the original work, including the introduction of new evaluation metrics and experimental setups for environments not covered by the original authors.

- **Extended Analysis:** By incorporating trajectory ranking within clusters, alternative trajectory clustering, and an expanded human study, the paper significantly builds upon the original findings, offering deeper insights into the explainability of RL decisions.

- **Quantitative and Qualitative Validation:** The study confirms the effectiveness of the original method through both quantitative and qualitative analysis, demonstrating the reproducibility of results across multiple environments and metrics.

- **Insightful Human Study:** The expanded human study provides valuable insights into the interpretability of the explanations generated by the method, highlighting the challenges faced by participants in more complex environments.

- **Environmental Impact Consideration:** The paper includes a section on computational requirements and carbon emissions, reflecting an awareness of the environmental impact of machine learning research.


## Weaknesses

- **Complexity in Interpretability:** The results suggest that while the method is effective in generating explanations, the complexity of those explanations increases with the complexity of the environment, potentially limiting its utility for end-users without domain expertise.

- **Implementation Challenges:** The paper discusses difficulties encountered due to missing version numbers and unclear calculation methods for certain metrics, which could have been mitigated with better documentation from the original authors.

- **Dependence on Clustering Quality:** The utility of the explanations seems highly dependent on the quality of trajectory clustering. While the paper explores alternative clustering methods, the dependence on this step underscores a potential fragility in the method's applicability across diverse RL tasks.

---

> ### Author Response · Authors · 2024-03-12
> **Reply to Reviewer 4piC**
>
> Thank you very much for your review. It is valuable to see what parts in our paper we could strengthen.

---

### Review · Reviewer_SnGL · 2024-03-06

**Summary Of Contributions:**

The manuscript attempts to reproduce the algorithms and experiments of "Explaining RL decisions with trajectories", a work that introduced a method for explaining RL policies learned using an offline dataset by comparing state action transitions with trajectories held in a training dataset. The authors expand that work by carrying out further human evaluation, and by doing some analysis on the trajectory clusters used in the method.

**Audience:**

Yes

**Claims And Evidence:**

Yes

**Requested Changes:**

What follows is a bunch of feedback and questions per page:

- p1: "While offline RL research..." -- I don't think offline RL as a framework has solved the issues you mention, and it's not sensible to imply that the list of RL-issues-in-real-life-scenarios is small. I can see what the manuscript is trying to say here, but adjustment is welcomed.
- p1: "...by highlighting critical state features" -- would be good to add a non-dense explanation here so that readers unfamiliar with the explanability literature is not forced into holding black boxes already. Same idea for what it means to "attribute" past experiences to agent decisions. Without a simple explanation here, the sentences don't have much unambiguous meaning (NB: things like the meaning of "trajectories" is instead well known in the RL community, so no need to clarify if you have verbosity constraints).
- p3: "...thus point to clusters of trajectories" -- I'd maybe rewrite this: "point to" is a little handwavy / confusing.
- p3: "...is trained on every cluster's complementary dataset" -- I would maybe point the reader to Figure 2d, as it's otherwise ambiguous.
- p4: "..why a cluster is attributed as responsible" -- reading the manuscript top down, it's not clear what this means. The paragraph feels also a bit roundabout-y about wanting to find the most similar trajectory in the selected cluster for analysis (because the hypothesis is that ultimately that's what matters the most here? Maybe worth clarifying the reason you are doing this analysis here?).
- p4: "A higher LMAAVD..." -- the explanation seems like a repetition of its previous sentence. Is it just trying to clarify the sign? Might be worth shortening.
- p5: "A lower NWD means..." -- why is it preferable?
- p6: "... because otherwise cluster labels would be shuffled" -- I understand why this is a problem in your analysis, but in your view if you had a way to fix this inconsistency, would you have been able to gain further insight in the results?
- p6: "... we experimented with different clustering algorithms..." -- this is only done in the grid-world setting. How do you think the higher state dimensionality for the other environments would affect the conclusion of generality wrt. clustering algorithms? Also, how did the new experiments pick a well functioning set of clusters? (e.g. a poorly trained / regularized set could massively influence later results)
- p6: "Human study" -- it would be helpful to have a single example of each type of question here, rather tha in the appendix.
- p6: "MachineLearning" -> "Machine Learning"
- p7: for section 4.1.2, I think it is necessary to qualify a little bit more how the inclusion sets work for each environment. Do the sets tend to have bottleneck states for instance, or any other clear structure?
- p8: "... NWD of 0": is this a problem with the experimental setting? Is the manuscript trying to argue that care is necessary when choosing clustering hyperparams or doing regularization? I would love to see some hypothesis (or better, detailed explanation) for why this happened.
- p9: "Anecdotal evidence" -- it would be great if the manuscript discussed any visible failure modes.
- p9: "... topologically and semantically similar to X-means" -- what's meant specifically here? Can you give examples of some that don't apply?
- p10: "..this is due to the inherent difficulty of understanding RL trajectories for humans" -- that's a somewhat surprising takeaway. Aren't these environments simple enough that good policies should be naturally interpretable? Is sub-optimality wrt. a clear reward function maybe the real factor?

**Strengths And Weaknesses:**

## Strengths

1. The manuscript is relatively easy to follow, and it is packed with experimental results.
2. I was able to understand the algorithm and generally the original work without having to look at its manuscript (even though later I did skim through that).
3. The manuscript's authors seem to have done a great job at being fair with the original work's authors, and have largely reproduced the original results.

## Weaknesses

My main concern with the paper is that the additional content on top of the original work is extremely limited. The paper fundamentally adds two things:

1. A way to rank trajectors against different explanation policy datasets -> I think this would be useful in the context of choosing hyperparams such as numbers of clusters and generally understanding the policy space better from a distributional perspective (e.g. bottleneck states, space variance). By itself, I'm not sure how much value it adds on top of the original work, and it's not clear to me whether we can make any strong claim from the data gained within the current experimental setting.
2. The human study extension -> I don't find the one major conclusion from the additional human experiments is not well defended: it's not clear to me whether these environments are "complex", and good ALE / simple mujoco environments are generally well interpretable in the literature, given how common it is to make qualitative analysis on top of policies trained in such environments. I would expect bad policies to be hard to interpret however, and there might be other factors at play here.

So, overall I find the manuscript useful, but I'm not sure about its added value outside of the efforts to reproduce the original work's findings.

Also, as a nit: I think the problem statement could be better introduced. See below.

---

> ### Author Response · Authors · 2024-03-11
> **Reply to the question of Reviewer SnGL**
>
> Dear Reviewer SnGL,
>
> Thanks you very much for the valuable feedback. We've rewritten the points in the requested changes and made sure they were less ambiguous. Additionally, we would like to answer your questions. Points that we have not explicitly listed here were edited in the manuscript.
>
> Question: p6: "... because otherwise cluster labels would be shuffled" -- I understand why this is a problem in your analysis, but in your view if you had a way to fix this inconsistency, would you have been able to gain further insight in the results?
>
> Answer: If there was a way for the clustering algorithm to consistently label the same groups of semantically similar trajectories with the same label, it might lead to more insight, as we could see the performance of distinct clusters over multiple cluserings. However, this is difficult to implement, as we have found no way to let the clustering algorithm do this consistently.
>
> Question: p6: "... we experimented with different clustering algorithms..." -- this is only done in the grid-world setting. How do you think the higher state dimensionality for the other environments would affect the conclusion of generality wrt. clustering algorithms? Also, how did the new experiments pick a well functioning set of clusters? (e.g. a poorly trained / regularized set could massively influence later results)
>
> Answer: We don't think that increased dimensionality would influence the performance of experimental clusterings to a point where the claim of the original authors would no longer hold (that any suitable clustering algorithm can be used instead of X-means). It is likely that experimental clusterings would be more unlike the original clustering than they were with Grid-World. However, we believe they would still be suitable for usage.
>
> Question: p8: "... NWD of 0": is this a problem with the experimental setting? Is the manuscript trying to argue that care is necessary when choosing clustering hyperparams or doing regularization? I would love to see some hypothesis (or better, detailed explanation) for why this happened.
>
> Answer: This is due to the normalization process from the original paper. We have added a clarification in the manuscript.
>
> Question: p10: "..this is due to the inherent difficulty of understanding RL trajectories for humans" -- that's a somewhat surprising takeaway. Aren't these environments simple enough that good policies should be naturally interpretable? Is sub-optimality wrt. a clear reward function maybe the real factor?
>
> Answer: This perspective is supported by the human study research conducted by Deshmuk et al. (2023), which suggests that "actual factors driving actions could be neglected by humans while understanding a decision", as per the human study results on questions type 2. In our human study, we aim to enhance participant understanding of decisions by utilizing gif images instead of a list of images. However, participants still express difficulty in comprehending trajectories within challenging environments, which is why we include these findings in our study.
>
> Once again, we are grateful for your insights. Please let us know if you have additional things you'd like us to clarify.

---

### Review · Reviewer_JUYP · 2024-03-07

**Summary Of Contributions:**

This paper is attempting to reproduce the results of a paper published at ICLR 2023, focused on explaining the behavior of an agent trained with a static dataset (offline reinforcement learning, RL). The original paper clustered trajectories and then explained decisions by finding the cluster that had the most impact on a particular decision. This paper additionally tries to rank particular trajectories within a cluster as well, extends human studies, and tests some alternative clustering techniques.

**Audience:**

Yes

**Claims And Evidence:**

Yes

**Requested Changes:**

In references, the reference for the main paper being reproduced, "Explaining RL decisions with trajectories", is not cited appropriately. No mention of where it was published is made.

**Strengths And Weaknesses:**

## Strengths
* The paper is attempting to reproduce the results of another paper. Thus, the exact settings it used to do so is important. The paper has gone into great detail about what the experimental setup and the hyperparameters used.
* It gives a clear overview of the paper it is trying to reproduce.
* The extension of the experimental results give additional value to the paper beyond just a validation of a previous paper
* The computational requirements and environmental impact section was particularly noteworthy
* The possible reasons for why reproducing the quantitative results was difficult was also appreciated

## Weaknesses
* The paper reproduces the results of the original paper in the same domains as the original paper. While this setup helps to validate if the original results can be reproduced, there is a question of how general the insights of the original paper are could also have been answered by evaluating on an additional domain.
* It is unclear what the scientific insight this study has provided. The additional experiments give us some deeper insight on the original paper. But it is unclear how valuable this insight is.
* Some of the results, such as the ones in Table 2, would benefit from clearer descriptions of what a reader should take away from it.
* In section 3.2, the first sentence could be expanded on to reduce murkiness and establish clearly what the algorithm is trying to do.

### Questions:
* In Section 4.2.1, what anecdotal evidence are the authors referring to? Their own?
* In section 4.2.2, how is the fact that the algorithm is not picking relevant trajectories a point confirming sub-claim 4? If I understand correctly, sub-claim 4 indicates that humans sometimes fail to correctly identify the correct factors influencing the RL decision. But Section  4.2.2 only indicates that humans do not pick some of the trajectories that the algorithm proposes. Perhaps the proposes trajectories are not relevant?

---

> ### Author Response · Authors · 2024-03-11
> **Reply to the questions of Reviewer JUYP**
>
> Dear Reviewer JUYP,
>
> We would like to thank you for your valuable feedback. We have implemented the requested changes in the manuscript. As an answer to your questions:
>
> Question: In Section 4.2.1, what anecdotal evidence are the authors referring to? Their own?
>
> Answer: This was ambiguous and we have changed it. We meant our own qualitative research.
>
> Question: In section 4.2.2, how is the fact that the algorithm is not picking relevant trajectories a point confirming sub-claim 4? If I understand correctly, sub-claim 4 indicates that humans sometimes fail to correctly identify the correct factors influencing the RL decision. But Section 4.2.2 only indicates that humans do not pick some of the trajectories that the algorithm proposes. Perhaps the proposes trajectories are not relevant?
>
> Answer: It is hard to say exactly what exactly makes a trajectory relevant as there is no ground truth. However, we believe that the method proposed by Deshmuk et al. (2023) proposes a robust way to find relevant trajectories. Therefore, we support the algorithm to find relevant trajectories. In that regard it would appear to us that it was the humans at error here, as they did not pick trajectories proposed by the algorithm. Of course, there is chance that this is incorrect. This area might benefit from further research.
>
> We hope this addresses any concerns you have. Please let us know if there is something we should elaborate on.

---

### Decision · Action_Editor_i3Hz · 2024-04-22

**Recommendation:** Accept as is

**Comment:**

The paper provides a reproducibility study, reexamine the results of a previously published paper, adds new experiments, and deepens the analysis of the original work.

(+) The paper is well-written.
(+) The methodology is reasonable. It validates the original findings and expands them in meaningful ways (although the expansion is limited).
(+) The changes made by the authors after the reviews addressed some of the reviewers' concerns.

(-) The new experiments are rather limited, and thus, the paper does not significantly expand our understanding of the original work.

**Audience:**

This paper could be of interest to those who are working on offline RL and explainable AI/RL.

**Claims And Evidence:**

The goal of the paper is to reproduce the algorithms and experiments of an existing ICLR-2023 paper entitled "Explaining RL decisions with trajectories". The ICLR-2023 paper proposes a method for explaining RL policies learned from an offline dataset by comparing state-action transitions with trajectories held in a training dataset. The authors expand that work by ranking particular trajectories within a cluster, carrying out human evaluation, and testing some alternative clustering techniques.